# Effects of Cinnamon Powder on Glucose Metabolism in Diabetic Mice and the Molecular Mechanisms

**DOI:** 10.3390/foods12203852

**Published:** 2023-10-20

**Authors:** Yaoyao Liu, Fan Liu, Dongxu Xing, Weifei Wang, Qiong Yang, Sentai Liao, Erna Li, Daorui Pang, Yuxiao Zou

**Affiliations:** 1Sericultural & Agri-Food Research Institute, Guangdong Academy of Agricultural Sciences, Key Laboratory of Functional Foods, Ministry of Agriculture and Rural Affairs, Guangdong Key Laboratory of Agricultural Products Processing, Guangzhou 510610, China; liuyaoyaov@163.com (Y.L.); liufan1234@126.com (F.L.); xingdongxu@gdaas.cn (D.X.); wangweifei@gdaas.cn (W.W.); yangqiong@gdaas.cn (Q.Y.); liaost@163.com (S.L.); snoopylen@126.com (E.L.); 2School of Food Science and Technology, Guangdong Ocean University, Zhanjiang 524088, China

**Keywords:** cinnamon powder, hypoglycemia, glycogen synthesis, gluconeogenesis

## Abstract

The liver is the primary organ regulating glucose metabolism. In our recent study, cinnamon improved liver function in diabetic mice. However, it is not clear whether cinnamon can reduce the glycemia of diabetic animals by regulating liver glucose metabolism. The purpose of this study was to investigate the hypoglycemic mechanism of cinnamon powder (CP) from the perspective of regulating liver glucose metabolism. To achieve this, different doses of CP (200, 400, or 800 mg/kg body weight) were given to diabetic mice by gavage once per day for 8 weeks. These mice were compared with healthy controls, untreated diabetic mice, and diabetic mice treated with metformin (the main first-line drug for type 2 diabetes). CP treatment effectively reduced fasting blood glucose levels and food intake, improved glucose tolerance and fasting serum insulin levels, and decreased glycated serum protein levels in diabetic mice. Furthermore, treatment with CP increased liver glycogen content and reduced the level of the gluconeogenesis precursor pyruvate in the liver. Data obtained by qPCR and western blotting suggested that CP improved glucose metabolism disorders by regulating AMPKα/PGC1α-mediated hepatic gluconeogenesis and PI3K/AKT-mediated hepatic glycogen synthesis. CP exhibits good hypoglycemic effects by improving hepatic glycogen synthesis and controlling hepatic gluconeogenesis. Therefore, CP may be applied as a functional food to decrease blood glucose.

## 1. Introduction

Cinnamon (*Cinnamomum cassia Presl*) is obtained from tropical evergreen trees belonging to the family Lauraceae. It is widely distributed in Sri Lanka, China, India, and Australia [1,2]. It has long been used as a spice and in traditional herbal medicines. Available evidence suggests that cinnamon has beneficial effects on diabetes management [3]. Multiple mechanisms have been proposed by which cinnamon improves diabetes, including enhancing insulin sensitivity through insulin receptor signaling, inhibiting the activity of carbohydrate-digesting enzymes, inhibiting glucose transport, delaying gastric emptying, and blocking glucose absorption [4,5,6]. The liver plays a crucial role in glucose metabolism. However, the effects of cinnamon powder (CP) on hepatic gluconeogenesis and glycogen synthesis have not been fully illustrated yet. Our recent study suggests that cinnamon improves liver function in diabetic mice by attenuating oxidative stress in liver tissue (Appendix A). Restoring liver function may further improve the regulation of genes and proteins involved in liver glucose metabolism signaling pathways and partially alleviate glucose metabolism disorders [7,8].

In this study, the hypoglycemic activity of CP was determined in vivo. Furthermore, the impact of CP on hepatic glucose metabolism was investigated. Additionally, the possible mechanisms of action involved in hepatic gluconeogenesis and hepatic glycogen synthesis of CP were explored.

## 2. Materials and Methods

### 2.1. Materials

STZ was supplied by Guangzhou Qiyun Biological Products Co., Ltd. (Guangzhou, Guangdong, China). Metformin was purchased from Sino-US Shanghai Bristol-Myers Squibb Pharma-Ceutical Co., Ltd. (Shanghai, China). Primary antibodies against peroxisome proliferator-activated receptor gamma coactivator 1-alpha (PGC-1a, cat. no. ab188102), glycogen synthase kinase 3 beta (GSK3β, cat. no. ab32391), glycogen synthase (GS, cat. no. ab40810), phosphorylated-AKT (p-AKT, cat. no. ab38449), and silent information regulator factor 2-related enzyme 1 (SIRT1, cat. no. ab189494) were supplied by Abcam (Cambridge, MA, USA). Antibodies against AMP-activated protein kinase alpha (AMPKα, cat. no. cst#5831), serine-threonine protein kinase (AKT, cat. no. cst#4691), and phosphorylated-AMPKα (p-AMPKα, cat. no. cst#2535) were supplied by Cell Signaling Technology (Danvers, MA, USA). Anti-GAPDH was acquired from Proteintech Group Inc. (Wuhan, Hubei, China). The other chemicals used in the experiment were all reagent-grade.

### 2.2. Sample Collection and Preparation

Fresh, matured bark of cinnamon was collected by the Guangxi Forestry Research Institute (Nanning, Guangxi, China). The bark of the plant was washed, air-dried, and ground into dry CP, which was stored at −20 °C until use.

### 2.3. Animals

Male BALB/c mice (4–6 weeks old) were obtained from the Laboratory Animal Center, Southern Medical University, Guangzhou, Guangdong Province, China (quality certification number: SYXK (Yue) 2020-0149). The animal experiments were approved by the Experimental Animal Ethics Committee of Sericulture and Agri-Food Research Institute, Guangdong Academy of Agricultural Sciences (29 January 2021, approval number: 2021-SC-03). Animals were acclimated to plastic-caged housing in a controlled environment (22 ± 2 °C and 12 h light/dark). Water and food were given ad libitum.

### 2.4. Induction of Experimental Diabetes

Diabetes was induced by the HFD/STZ method [9]. After the adaptation time, except in the blank control (normal) group, the mice were fed a HFD (60% energy as fat) for 4 weeks. The mice were fasted overnight before induction of diabetes and then intraperitoneally injected with STZ (80 mg/kg body weight (b.w.)) dissolved in citrate buffer (0.1 mol/L, pH 4.5). After 7 days, fasting blood glucose (FBG) levels were estimated with a glucometer (ACCU-CHEK Active Blood Glucose Meter, Roche) by drawing blood from the tail vein. The diabetic model that produces FBG > 11.1 mmol/L is considered successful [10].

### 2.5. Experimental Design

Diabetic mice were randomized into five groups (10 mice per group): Model group (diabetic mice not treated with a drug): orally administered vehicle (saline) once per day; metformin group: diabetic mice orally administered metformin at 200 mg/kg b.w. once per day; CP-200 group: diabetic mice orally administered CP at 200 mg/kg b.w. once per day; CP-400 group: diabetic mice orally administered CP at 400 mg/kg b.w. once per day; CP-800 group: diabetic mice orally administered CP at 800 mg/kg b.w. once per day; normal group (healthy mice): orally administered vehicle (saline) once per day (10 mice in this group). Body weight, food intake, and FBG were monitored each week. After 8 weeks of gavage, all of the mice were euthanized by carbon dioxide inhalation. Blood was centrifuged, collected, and stored at −80 °C. The livers were rapidly collected, weighed, and separated into three specimens. Liver tissues were immediately frozen at −80 °C until analysis.

### 2.6. Oral Glucose Tolerance Test (OGTT)

An OGTT was carried out following the method described by Xiao et al. [11]. The test was performed in week 8, after the mice had fasted overnight for 12 h. Subsequently, a glucose load of 1 g/kg body weight was administered. Blood glucose levels were monitored at 0, 30, 60, and 120 min by tail puncturing. The area under the glycemic curve (AUC) was measured using the following formula:$$\mathrm{AUC}=\frac{0.5A+B+C+0.5D}{2}$$
where A, B, C, and D represent the blood glucose values at 0 min, 30 min, 60 min, and 120 min, respectively, after oral administration of the glucose solution.

### 2.7. Biochemical Assays in Serum and Liver

All liver and serum biochemical indicators were determined using kits (Nanjing Jiancheng Bioengineering Institute, Nanjing, Jiangsu, China). Serum insulin concentration was detected by enzyme-linked immunosorbent assay (Qiaoyi Biological Technology Co., Ltd., Hefei, Anhui, China).

### 2.8. RNA Preparation and Quantitative Real-Time RT-PCR (qPCR)

For the qPCR analysis, the methods described by Chen et al. were followed with slight modifications [12]. Total RNA was extracted from liver tissue using the Trizol reagent (Gibco, Rockville, MD, USA), followed by reverse transcription into cDNA using a HiScript 1st Strand cDNA Synthesis Kit. Real-time PCR assays were conducted using a SYBR FAST qPCR Master Kit (KAPA, cat. no. KK4610) according to the provided instructions. Relative mRNA expression levels were normalized using GAPDH as the housekeeping gene. The data were analyzed using the 2^−ΔΔCt^ method. The primer sequences used in qPCR are listed in Table 1.

### 2.9. Western Blotting

Western blotting was carried out as detailed previously [12]. Briefly, protein lysates obtained from liver homogenate were electrophoresed, blotted, and then incubated with antibodies against PI3K, AKT, GSK3β, GS, AMPKα, SIRT-1, and PGC1α, with appropriate secondary antibodies. GAPDH was used as an internal control.

### 2.10. Data Analysis

Data are presented as the mean ± SD. SPSS Statistics software v21.0 (SPSS Inc., Chicago, IL, USA) was used to conduct one-way analysis of variance and Duncan’s multiple range test. A calculated *p*-value < 0.05 was considered statistically significant.

## 3. Results

### 3.1. Effects of CP on Body Weight and Food Intake in Diabetic Mice

The effects of CP on b.w. and food intake in mice are shown in Figure 1. Before drug administration, there was no statistical difference in b.w. between the mice in different groups. After 8 weeks of drug administration, CP-400 (29.11 ± 2.49 g), CP-800 (28.88 ± 1.97g), or metformin (30.59 ± 0.64 g) treatment significantly suppressed the drop in b.w. seen in the model group (25.81 ± 2.23 g) (Figure 1A). At the start of the treatment, all diabetic mice consumed more food than normal mice (*p* < 0.05, Figure 1B). After being treated with CP for 8 weeks, the food intake of mice in the CP-200, CP-400, and CP-800 groups was 7.09 ± 0.68 g/day, 6.75 ± 0.31 g/day, and 5.14 ± 0.53 g/day, respectively. This represents a reduction of 21.4%, 25.1%, and 43.0% compared to the model group. These results suggest that treatment with CP significantly affected the b.w. and food intake of diabetic mice.

### 3.2. Effects of CP on FBG, Glycosylated Serum Protein (GSP), and Fasting Serum Insulin (FINS) Levels in Diabetic Mice

After administration of CP for 8 weeks, FBG levels prominently increased in the model group (27.35 ± 3.87 mmol/L) compared to the normal group (4.00 ± 0.57 mmol/L, *p* < 0.05), suggesting that diabetes was successfully induced. The FBG of mice in the CP-200, CP-400, and CP-800 groups was 23.06 ± 1.65 mmol/L, 23.20 ± 3.45 mmol/L, and 17.83 ± 3.67 mmol/L, respectively. This shows a decrease of 15.7%, 15.2%, and 34.8% compared to the model group. These findings indicate that CP-800 had a relatively high hypoglycemic effect (Figure 2A). In Figure 2B, it can be observed that the mice in the normal group had the highest serum GSP levels, approximately 6.78 ± 0.50 mmol/L. The serum GSP levels of the model group were the lowest, only 3.90 ± 0.74 mmol/L, which was significantly different from the normal group (*p* < 0.05). The groups receiving CP-800 intervention showed significantly reduced serum levels of GSP compared to the model group (*p* < 0.05; Figure 2B). Compared to the normal mice (0.16 ± 0.06 ng/mL), the FINS levels decreased significantly in the model group (0.06 ± 0.02 ng/mL, *p* < 0.05, Figure 2C). However, the groups receiving CP-800 intervention showed a marked improvement in FINS levels compared to the model group (*p* < 0.05).

### 3.3. Effects of CP on OGTT and AUC

The OGTT is an important index for evaluating bodily regulation and tolerance of glucose [10]. To confirm the hypoglycemic effects of CP, glucose tolerance experiments were performed on mice in different groups. The blood glucose levels of mice in all treatments first increased and subsequently dropped during the 120 min detection period (Figure 3A). Groups receiving CP or metformin interventions showed an obviously inhibited rise in blood glucose levels after oral glucose administration compared with the model group. The glucose excursion was quantified as the AUC during the 120 min test (Figure 3B). The AUC was significantly lower in the CP-200 (53.98 ± 4.66 mmol/(L·h)), CP-400 (50.84 ± 3.84 mmol/(L·h)), and CP-800 (49.68±4.34 mmol/(L·h)) groups than in the model group (58.63 ± 3.92 mmol/(L·h)) (*p* < 0.05), and the reduction rates were 7.9%, 13.3%, and 15.3%, respectively. Thus, medium and high doses of CP had a marked effect on oral glucose tolerance, although no significant differences were observed. To summarize, we demonstrate that CP can lower blood sugar and improve glucose intolerance in diabetic mice.

### 3.4. Effects of CP on the Glycogen and Pyruvate Levels in the Liver of Diabetic Mice

The liver is the primary organ regulating glucose metabolism [13]. Recently, a study in our laboratory demonstrated that cinnamon could improve liver function in diabetic mice by attenuating oxidative stress in liver tissue. However, it is not clear whether cinnamon can moderate glucose metabolism in liver tissue. Therefore, we determined the glycogen and pyruvate levels in the livers of diabetic mice in the various experimental groups. As indicated in Figure 4A, the content of hepatic glycogen in the model and normal groups was 0.05 ± 0.01 mg/g and 0.13 ± 0.01 mg/g, respectively. Compared to that in normal mice, the content of hepatic glycogen in model mice was reduced significantly by 58.2% (*p* < 0.05). This demonstrates that liver glycogen synthesis or storage may be lessened in diabetic mice. After administration of CP for 8 weeks, the hepatic glycogen content in the CP-200, CP-400, and CP-800 groups was 0.068 ± 0.005 mg/g, 0.089 ± 0.016 mg/g, and 0.111 ± 0.007 mg/g, respectively. The CP-800 and CP-400 groups had higher liver glycogen content than the model group (*p* < 0.05). This result demonstrates that CP can enhance liver glycogen synthesis and/or storage in diabetic mice.

After administration of CP for 8 weeks, the level of pyruvate in the normal and model groups was 0.012 ± 0.001 μmol/mL and 0.132 ± 0.022 μmol/mL, respectively. Compared with that in normal mice, the level of pyruvate in the model group increased 7.4-fold (*p* < 0.05, Figure 4B), suggesting exaggerated gluconeogenesis in diabetic mice. CP treatment significantly reduced hepatic pyruvate content in a dose-dependent manner when compared with the model group. The hepatic pyruvate content in the high-dose (CP-800) group decreased by 74.7%. These results indicate that CP intervention reduces levels of pyruvate, a precursor of liver gluconeogenesis.

### 3.5. Effects of CP on Relative Hepatic Expression of Genes Involved in Glycogenesis and Gluconeogenesis

To explore the mechanism(s) by which CP affects glycogen synthesis and gluconeogenesis in the liver, the expression of mRNAs related to glycogen synthesis and gluconeogenesis was analyzed by qPCR experiments. The diabetic control (model) group showed significantly impaired insulin signaling, including decreases in *pi3k* and *gs* transcript levels, as well as increases in the transcript levels of *gsk3β*, compared with the normal group. In comparison with the model group, CP-800 intervention significantly improved the transcript levels of *pi3k* and *gs* and suppressed the transcript levels of *gsk3β* (*p* < 0.05; Figure 5A).

As shown in Figure 5B, the transcript levels of *sirt-1* and *pgc1-α* significantly decreased in model mice compared to normal mice (*p* < 0.05). Compared with the model group, the groups receiving CP-400 and CP-800 intervention showed markedly increased transcript levels of *sirt-1* and *pgc1-α*; in the CP-800 group, the transcript levels of *sirt-1* and *pgc1-α* were increased 57.7-fold and 7.0-fold, respectively. These results suggest that CP may increase the transcript levels of genes encoding proteins involved in hepatic gluconeogenesis.

### 3.6. Effects of CP on Relative Hepatic Expression of Proteins Involved in Glycogenesis and Gluconeogenesis

To confirm the results obtained from the analysis of mRNA (i.e., gene expression) levels, western blotting was used to detect and quantify PI3K/AKT and AMPK/PGC1α pathway-related protein levels in liver tissue. In comparison with the model group, both CP-400 and CP-800 interventions significantly increased p-AKT protein expression (*p* < 0.05, Figure 6A). In comparison with the normal group, the diabetic control (model) group showed significantly increased GSK3β protein levels and decreased GS-protein levels (*p* < 0.05, Figure 6B,C). After CP-800 intervention, the GSK3β protein expression level significantly decreased, while the GS-protein expression level significantly increased compared with the model group (*p* < 0.05). The above results indicate that CP regulated the relative protein expression levels of PI3K/AKT/GSK3β/GS in the liver of diabetic mice, resulting in an increase in hepatic glycogen synthesis.

In comparison with the normal group, the diabetic control (model) group showed significant reductions in AMPKα, SIRT-1, and PGC1-α proteins. When compared with the model group, CP markedly increased the relative expression of p-AMPK, SIRT-1, and PGC1-α; the SIRT-1 and PGC1-α protein levels in the CP-800 group increased 8.6-fold and 0.4-fold, respectively. The p-AMPK/AMPK ratio in the CP-800 group increased 1.9-fold. Thus, CP may moderate gluconeogenesis in diabetic mice by modulating the relative expression levels of proteins that are involved in hepatic gluconeogenesis.

## 4. Discussion

The results of the previous study indicate that cinnamon is a potential food raw material that can be consumed to improve glucose metabolism [14]. This effect may be due in part to its polyphenol composition [15]. We previously analyzed cinnamon polyphenol compounds using ultra-high-performance liquid chromatography-tandem mass spectrometry. A total of 145 phenolic compounds were identified in CP, with the main components being procyanidin B3 (3262.66 ± 176.82 μg/mL dry weight), procyanidin B2 (2592.68 ± 33.84 μg/mL dry weight), procyanidin A2 (417.47 ± 17.80 μg/mL dry weight), B1 (282.26 ± 12.44 μg/mL dry weight), procyanidin C1 (112.59 ± 10.78 μg/mL dry weight), (−)-epicatechin (795.53 ± 68.80 μg/mL dry weight), (−)-catechin (339.29 ± 35.01 μg/mL dry weight), quercitrin (182.81 ± 5.69 μg/mL dry weight), theaflavin (124.87 ± 4.97 μg/mL dry weight), kaempferitrin (106.45 ± 1.35 μg/mL dry weight), and avicularin (100.96 ± 5.84 μg/mL dry weight). The highest concentration was procyanidin B3, followed by procyanidin B2 and (−)-epicatechin [16]. Previous animal studies have suggested that procyanidin oligomers may be responsible for the antidiabetic activity of cinnamon [17]. Cinnamon-derived A- and B-type procyanidin oligomers have hypoglycemic activity and may improve liver glucose metabolism in type 2 diabetes [18]. Furthermore, cinnamon may have hypoglycemic effects in part because of the presence of dietary fiber. Dietary fiber can control the release of glucose in the blood over time, thus contributing to the control and management of diabetes [19]. However, the health benefits of foods are likely to result from the additive and synergistic effects of different kinds of phytochemicals rather than a single component [20]. Previous studies have also suggested that there are synergistic effects of dietary fiber and polyphenols on antidiabetes [21,22,23]. These findings suggest that the additive and synergistic effects of these phytochemicals in CP may be the major reason for its antidiabetic effect.

Pharmacologically targeting signaling molecules to mediate gene or protein expression is an attractive concept to prevent diabetes. AKT, a serine/threonine protein kinase, is a major downstream effector of PI3K [24]. Activated AKT promotes glycogen synthesis and glycolysis through a variety of downstream targets [25]. When the blood sugar level increases, insulin stimulates the activation and phosphorylation of PI3K and AKT, thereby regulating downstream GSK3β and GS, which are the main regulatory factors for glycogen synthesis [26]. This is consistent with the changes in CP regulation-related gene expression in the PI3K/AKT pathway observed in this study. The changes in gene expression were essentially consistent with the changes we observed in protein expression levels. CP regulates the phosphorylation of AKT by activating PI3K, thereby inhibiting GSK3β and GS downstream signaling and ultimately increasing liver glycogen synthesis (Figure 7).

PGC-1-α is a multifunctional protein and a key regulator of the gluconeogenesis pathway. It is activated via phosphorylation by AMPK and subsequent deacetylation by SIRT-1 for activation of gluconeogenesis [27]. Our data suggest that CP can regulate the transcriptional and translational levels related to gluconeogenesis via the AMPKα/PGC1-α pathway in diabetic mice. CP activates AMPKα phosphorylation to promote the expression of SIRT-1 and PGC-1α, thereby reducing liver gluconeogenesis (Figure 7). In addition, higher pyruvate levels can drive hepatic gluconeogenesis [28]. CP intervention decreased liver pyruvate levels, consistent with results suggested by analysis of signal transduction processes. Taken together, these results indicate that CP exhibits good hypoglycemic effects by controlling hepatic gluconeogenesis.

The changes observed in glucose metabolism-related targets at the transcription level are consistent with those at the translation level. However, the regulatory effect on gene expressions of CP is more obvious than that of protein expressions. These results may be attributed to the regulation of mRNA by RNA interference, such as endogenous microRNAs (miRNAs) [29,30,31]. Research has shown that hepatic miRNAs play important roles in regulating the signaling pathways of liver glycogen synthesis and hepatic gluconeogenesis [32]. miR-378, miR-107, miR-103, miR-143, and miR-19a, miR-153, among others, may be involved in the regulation of the PI3K/AKT pathway [29,33]. miR-34a-5p and miR-696 may be involved in the regulation of the AMPKα/PGC1α pathway [34,35]. These mRNAs may affect the signaling processes of liver gluconeogenesis and liver glycogen synthesis.

A previous study reported that cinnamon extract exhibits a hypoglycemic effect in HFD-fed rats by upregulating the IRS1/PI3K/AKT2 signaling pathway [36]. This result suggests that cinnamon extract regulates the signaling pathway of liver glycogen synthesis, which is consistent with our study. Additionally, GLUT4 is the main transporter responsible for removing most of the glucose from the circulation after insulin stimulation. Previous studies have demonstrated that cinnamon extract upregulates the expression of GLUT4 in the cytoplasmic membrane [37,38,39]. However, our research yielded different results. These differences may be attributed to the different active ingredients in cinnamon extract and CP. In this study, CP, at a dose of 800 mg/kg bw, exhibited significant hypoglycemic activity comparable to that of metformin. A previous study suggested that the effective hypoglycemic dose for CP in mice is approximately 300 mg/kg b.w. [40]. The effective dosage of cinnamon can vary depending on factors such as the variety of cinnamon, which can influence the bioactive contents of the plant material and consequently affect its bioactivity [6]. Additionally, the effective dosage and intervention duration may also be influenced by the duration of treatment and the route of administration.

## 5. Conclusions

CP has shown potential for reducing hyperglycemia in diabetic mice. The mechanism by which CP regulates hepatic glucose metabolism is through control of liver gluconeogenesis and improvement of liver glycogen synthesis by activating P13K/AKT and AMPKα/PGC1α signaling pathways, resulting in a significant hypoglycemic effect. Further studies on the action mechanism of CP have confirmed that cinnamon exerts its glucose-lowering effect partly by promoting the conversion of glucose to glycogen and other non-sugar substances. Overall, CP may be an effective and cost-efficient dietary or therapeutic agent for the treatment of type 2 diabetes. Future studies should investigate the potential additive or synergistic effect of different components in CP and further evaluate the RNA interference effect of small RNA in regulating the signal pathways of liver gluconeogenesis and liver glycogen synthesis after the administration of CP.

## Figures and Tables

**Figure 1 foods-12-03852-f001:**
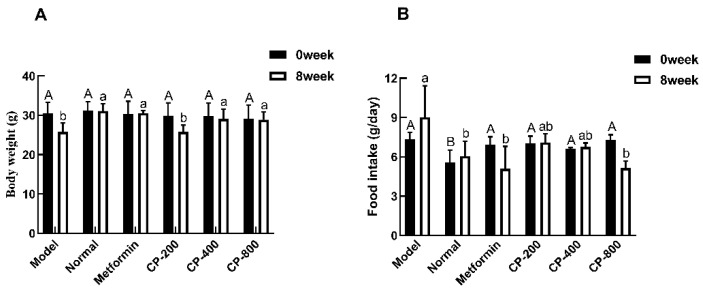
Effects of cinnamon powder (CP) treatment on the body weight (**A**) and food intake (**B**) of experimental mice. The data are the mean ± SD (*n* = 10). Bars marked with different letters of the same case are significantly different (*p* < 0.05).

**Figure 2 foods-12-03852-f002:**
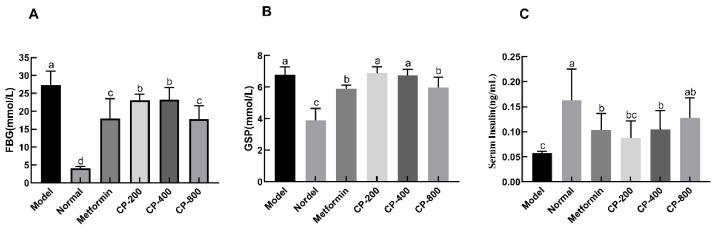
Effects of CP treatment on fasting blood glucose (FBG) (**A**), glycosylated serum protein (GSP) (**B**), and fasting serum insulin (FINS) (**C**) levels in diabetic mice. The data are the mean ± SD (*n* = 10). Bars marked with different letters are significantly different (*p* < 0.05).

**Figure 3 foods-12-03852-f003:**
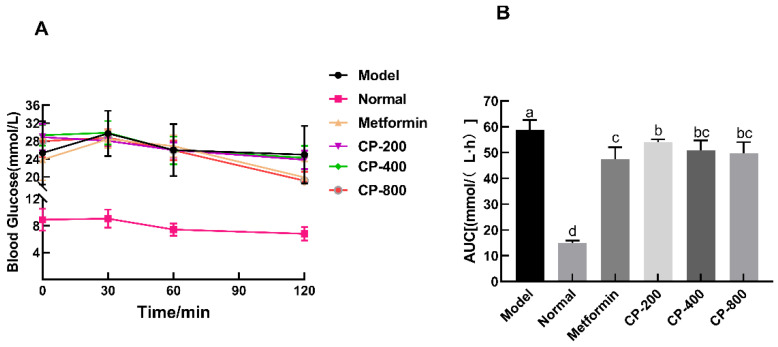
Effects of CP treatment on the oral glucose tolerance test (**A**) and on the area under the glycemic curve (AUC) (**B**). The data are the mean ± SD (*n* = 10). Bars marked with different letters are significantly different (*p* < 0.05).

**Figure 4 foods-12-03852-f004:**
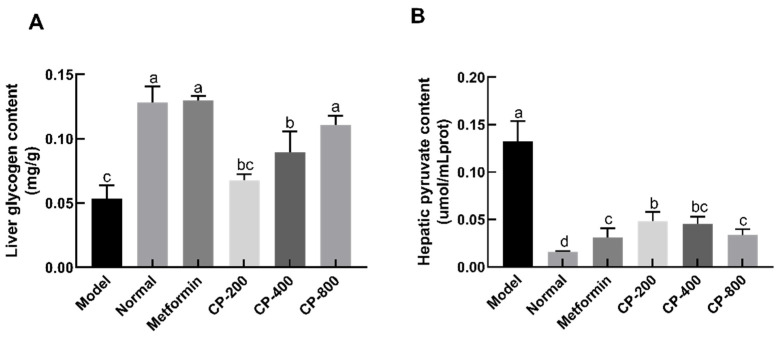
Effects of CP on glycogen (**A**) and pyruvate levels (**B**) in the liver of diabetic mice. The data are the mean ± SD (*n* = 10). Bars marked with different letters are significantly different (*p* < 0.05).

**Figure 5 foods-12-03852-f005:**
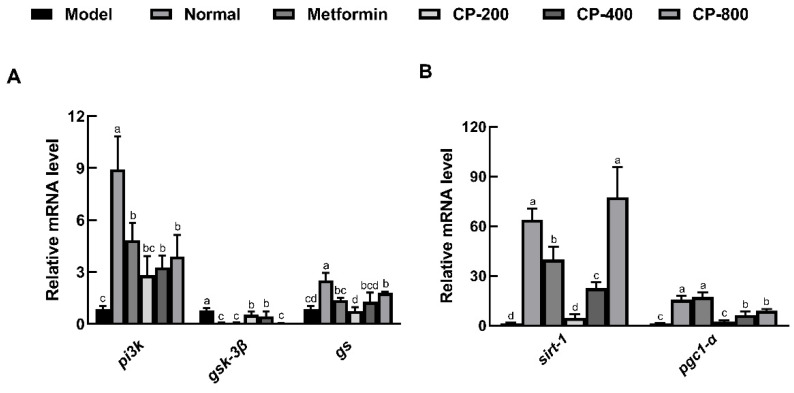
Effects of CP on the relative expression of genes related to liver glycogen synthesis and gluconeogenesis in diabetic mice. Gene expression related to the PI3K/AKT pathway (**A**); gene expression related to the AMPK/PGC1-α pathway (**B**). The data are the mean ± SD (*n* = 10). Bars marked with different letters are significantly different (*p* < 0.05).

**Figure 6 foods-12-03852-f006:**
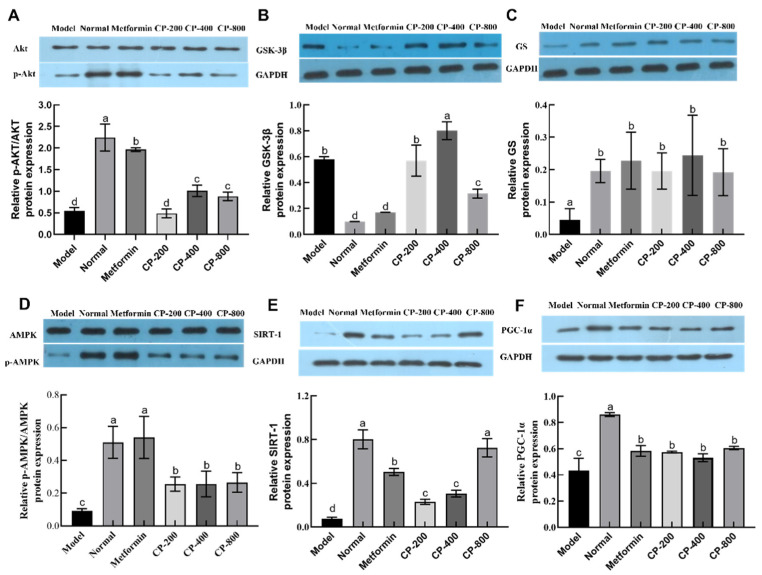
Effects of CP on protein expression in the PI3K/AKT and AMPK/PGC-1α pathways in the livers of diabetic mice. (**A**) p-AKT/AKT; (**B**) GSK-3β; (**C**) glycogen synthase (GS); (**D**) p-AMPK/AMPK; (**E**) SIRT-1; (**F**) PGC-1α. The data are the mean ± SD (*n* = 10). Bars marked with different letters are significantly different (*p* < 0.05).

**Figure 7 foods-12-03852-f007:**
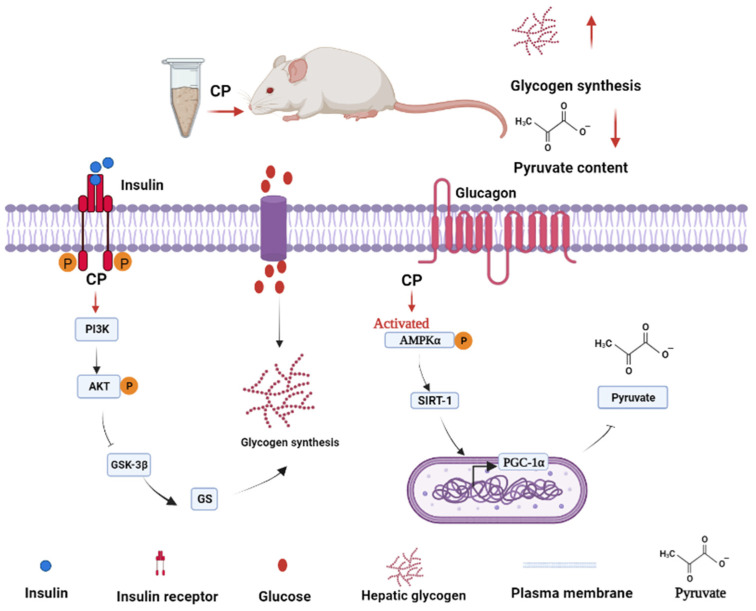
Possible mechanisms of the effect of CP on hepatic glucose metabolism in diabetic mice.

**Table 1 foods-12-03852-t001:** Primer sequences used in RT-qPCR.

Gene	Forward Primer (5′-3′)	Reverse Primer (5′-3′)
*gapdh*	GGAGAAACCTGCCAAGTATGATGAC	GAGACAACCTGGTCCTCAGTGTA
*pi3k*	TATGCCTGCTCCGTAGTGGTAGAC	GGTGGCTTGAAGGTGAGGAACTG
*akt*	CACCGTGTGACCATGAACGAGTT	TGGCGACGATGACCTCCTTCTT
*gsk3β*	TAATGCTGGAGACCGTGGACAG	CGTGACCAGTGTTGCTGAGTG
*gs*	GAACAGACGGCCACCCATT	CACTGGGCAGGCATAACCT
*ampkα*	CGTCGCCTACCACCTCATCATAGA	TCGGCAACCAAGAACGGTACTCT
*sirt-1*	GTGGCAGTAACAGTGACAGTGGC	TCCAGATCCTCCAGCACATTCGG
*pgc1α*	ATGTGTCGCCTTCTTGCTCTTCC	CGGTGTCTGTAGTGGCTTGATTCAT

## Data Availability

The data used to support the findings of this study can be made available by the corresponding author upon request.

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
