# Peer review of "Effects of Cinnamon Powder on Glucose Metabolism in Diabetic Mice and the Molecular Mechanisms"

_foods, 2023, doi:10.3390/foods12203852_

Round 1

Reviewer 1 Report

Comments and Suggestions for Authors

Dear Authors

The manuscript (foods-2637026) entitled “Effects of cinnamon powder on glucose metabolism in diabetic mice and the molecular mechanisms” is well written, has an important scientific message and should be of great interest to the readers of Food journal (ISSN 2304-8158) (Section: Food Nutrition; Intervention Effect of Natural Food Products on Chronic Diseases - Volume II).

The manuscript presents interesting and scientific important results. The authors investigate the hypoglycemic mechanism of cinnamon powder from the perspective of regulation of liver glucose metabolism. They found that cinnamon powder exhibits good hypoglycemic effects by improving hepatic glycogen synthesis, as well as controlling hepatic gluconeogenesis. Therefore, cinnamon powder may be applied in functional foods to decrease blood glucose.

- Abstract is comprehensive by itself. The important and essential information of the article is included

- References: appropriate and adequate references to related works are covered sufficiently in the list

- Structure and length: the overall structure of the article is well organized and well balanced. The article is written with the minimum length necessary for all relevant information.

- Novelty and originality: the article is novel and original. The article contains material that is new or adds significantly to knowledge already published.

The manuscript presents interesting and clinically important results. A few issues, however, need to be addressed:

*** Please insert a list of abbreviation after the Abstract section.

*** Introduction section

- Failure to place the study in a broad context

- Introduction that does not establish the background of the problem studied. The introductory section does not adequately explain the framework and problems of the research.

Author Response

请参阅附件。

Reviewer 2 Report

Comments and Suggestions for Authors

The manuscript is very interesting. The manuscript is written very carefully, and its subject fits in the scope of the Foods journal. The authors have divided the text into several chapters. The division of the text into subsections is appropriate.

Introduction: presents the topic in an appropriate way; However, I believe that the authors should refer to hepatic expression of proteins involved in glycogenesis

Purpose of the study: well formulated; the aim of the study was to investigate the effect of powdered cinnamon as a dietary supplement on the regulation of glucose levels in the liver and the possible mechanisms of action related to the regulation of glucose levels in the liver.

Materials and methods: described in great detail; I do not have any comments or suggestions

Results: I do not have any comments or suggestions  

Discussion: The effect of cinnamon powder on protein expression in the liver should be analyzed; Can the amount of cinnamon used affect the result?

Conclusions: I do not have any comments or suggestions

References: I believe that relevant literature was used in the study to explore the issues fully.

Tables and figures: I do not have any comments or suggestions

Reviewer 3 Report

Comments and Suggestions for Authors

Effects of cinnamon powder on glucose metabolism in diabetic 2 mice and its effect on the liver function and possible mechanisms were reported in this manuscript. It is a very well designee and written manuscript. Introduction part, method section and results section were written and presented very well.

 It could be better to present a Table on the composition of cinnamon powder used in this study as in the discussion section it was referred to this data.

Just one point should be clarified or may be corrected, line 249-250, the dietary fiber present in the cinnamon powder stated as a reason for the function of CP. As it is clear from the study design, the amount of cinnamon powder (as medicine) used in this study and as a consequence its fiber content is not that level which can have significant effect.

Author Response

请参阅附件。

Reviewer 4 Report

Comments and Suggestions for Authors

 Yaoyao Liu et al studied on Effects of cinnamon powder on glucose metabolism in diabetic 2 mice and the molecular mechanisms. The observation on the manuscript is as follows:

1.      The title seems befitting

2.      Please see the annotated comments in the abstract section

3.      The introduction looks misleading. Authors did not use the supplement form of Cinnamon ((Fresh matured bark of cinnamon ) but they have focused its food supplemental use in the objective.

4.      Section 2.2-what was the body weight of animals?

5.      Authors must incorporate the ethical protocol number

6.      Authors used the STZ in a very high dose ([80 mg/(kg·body weight (b.w.))]. Without reference it is not accepted. Well accepted dose ranges from 45-60 mg/kg

7.      Glucometer needs to be specified because it does matter (to my experience)

8.      Section 2.5------the CP at 400 and 800 are too high doses. Especially the 800 mg/kg is not acceptable. How did you administrate such a high dose to the mice? IT s not feasible. And authors did not explain any basis of using these doses

9.      Section 2.6------When the OGTT was conducted? The information is not sufficient. Based on what OGTT was decided

10.   Tables --------1; how did you choose thse genes? There are more genes responsible for metabolic actions

11.   Section 3.1-----------------Diabetes affect both food and fluid intake, why you did not consider fluid intake? It must be addressed for evaluating the metabolic action

12.   Figure 2C---------Why the error bars are so long? IT seems the authors have used the animals unequally or the data optimization was not sound

13.   Figure 3A is too clumsy, needs to use Xcel rather than GraphPAd prism,,I think its drawn by GraphPAd prism.By the way, it needs to redraw

14.   Line 180-182, Its result, no need to draw conclusion with interpretation

15.   Figure 5 needs to be converted into table because the data are not clearly shown

16.   Section 4………………without discussion, how are you concluding? Rephrase the discussion. The whole discussion is written based on previous study not current study. Authors need to rigorously revise the discussion

17.   In the conclusion, how can you say “CP has the potential to reduce hyperglycemia in diabetic mice. Polyphenols, such as 277 procyanidin B2, C1, B3, A2, and others, may be among the active substances in CP”-----hve you studied all these in this study? Flaws-----------

18.   The current conclusion is not acceptable

Comments on the Quality of English Language

Round 2

Reviewer 2 Report

Comments and Suggestions for Authors

The manuscript is very interesting and fits in the scope of the Foods journal.

I have thoroughly reviewed the manuscript. I believe the authors addressed all the issues raised by the reviewer satisfactorily.  I recommend this manuscript for publication.

Reviewer 4 Report

Comments and Suggestions for Authors

Authors addressed the suggestions except converting the Figure 5 to table